# Usefulness of a Drug Information Resources Website (SAGASU-DI) Created Based on Inquiries to Clinical Pharmacists

**DOI:** 10.3390/healthcare10081541

**Published:** 2022-08-14

**Authors:** Toshiki Murasaka, Hideo Kato, Hirofumi Sudo, Takuya Iwamoto

**Affiliations:** 1Department of Clinical Pharmaceutics, Mie University Graduate School of Medicine, Mie University, Tsu 514-8507, Japan; 2Konan Pharmacy, Tsu 514-0315, Japan; 3Department of Pharmacy, Mie University Hospital, Tsu 514-8507, Japan

**Keywords:** drug information, website, community pharmacist, Google Analytics

## Abstract

The use of drug information is increasing as the role of pharmacists expands. However, pharmacists spend a huge amount of time collecting drug information, and there has not been any drug information resource website that aims to improve the efficiency of community pharmacists with regard to drug information operations. The purpose of this study was to evaluate the usefulness of a drug information resources website (SAGASU-DI). SAGASU-DI articles were created based on inquiries to clinical pharmacists. The usage statistics of the SAGASU-DI were monitored using Google Analytics between May and July 2021. In addition, a pop-up questionnaire was used to evaluate the usefulness of the SAGASU-DI in answering the questions of viewers. Statistics showed that our website had 25,447 users and 33,000 page views, with a browsing time of 29 s on average. Visitors accessed the website through desktop (51.9%) and mobile (44.3%) systems. Community pharmacists accounted for 40.2% of the visitors. The pop-up questionnaire showed that 23.2% of the viewers were satisfied with our website, and 1.5% of the viewers were not. Although 65.1% of the viewers refused to answer the questionnaire, the average percentage of the viewers who were satisfied with the site was 94.1%. The SAGASU-DI was found to be a valuable resource website for drug information services, mainly for community pharmacists.

## 1. Introduction

Pharmacists need to obtain appropriate drug information (DI) to help provide counsel to patients and physicians in clinical settings and form a foundation for pharmaceutical care that ensures appropriate medication use [1,2]. Therefore, pharmacists need to use DI resources such as accurate, latest, and credible articles.

Recently, medical knowledge has substantially increased, and the way to access DI has changed notably over the last few decades [3]. Printed resources are rapidly being replaced by online resources and mobile applications [3]. Moreover, Bates advocated for web-based DI resources to be at the forefront of the healthcare field owing to their value in enhancing safety and improving patient outcomes [4]. Indeed, 70% of the pharmacists who practice in both hospital and community settings responded that the DI resource most frequently used in their pharmacy was electronic [5]. This finding indicates that electronic resources often provide evidence that can be used to quickly resolve queries.

Traditionally, the main role of community pharmacists has been to prepare medications based on prescriptions after checking dosage and administration, drug and food interactions, and side effects [6,7]. Currently, the Japanese government is promoting a “community-based integrated care system” and a “family pharmacist system” [7,8] and urging the community pharmacists to change their services from product-centered care to patient-centered care. However, community pharmacists have little time to collect DI because they need to perform patient-centered care in addition to the various services that they usually perform [9].

Google Analytics is an open service launched by Google in November 2005 and provides free quantitative data on website usage. This tool is designed to understand targets and their needs from a marketing perspective. To date, Google Analytics has also been used in health research as a process evaluation of web-based interventions using numerous variables related to webpage traffic [10]. To our knowledge, there has not been any DI resource website that aims to improve the efficiency of community pharmacists in carrying out DI operations, and the application of Google Analytics for the evaluation of web-based DI has not yet been reported.

Therefore, in this study, we built and launched a DI website (SAGASU-DI) for community pharmacists on the Internet. Currently, the total number of views on the SAGASU-DI is over 1.5 million.

The purpose of our study was to evaluate the usefulness of the SAGASU-DI as a DI resource website developed by community pharmacists with experience in DI operations in hospitals using Google Analytics and questionnaires to verify whether the site is useful for DI work.

## 2. Materials and Methods

Design and setting: In this study, we investigated the web traffic of the SAGASU-DI for three months (1 May–31 July 2021). The study population included all users on the website. The website was evaluated using Google Analytics data and a pop-up questionnaire. The SAGASU-DI was created by Konan Pharmacy (Mie, Japan) in October 2016 and has been continually updated by adding new documents on DI. It is a freely accessible web (https://sagasudi.com accessed on 1 May 2021) with 616 documents in Japanese and English. The documents in Japanese can be translated to English using the automatic translation function implemented on the website. The documents were managed by a pharmacist with 3 years of experience as a hospital pharmacist and 10 years of experience as a community pharmacist. The questions of Q&A documents on the website were created based on inquiries about the drug information (DI) received by Konan Pharmacy from doctors, pharmacists, nurses, nursing home staff, and patients. The answers in Q&A documents were created based on reliable information sources such as “prescribing information”, which are official documents in accordance with the Pharmaceutical Affairs Law, and the interview forms, which are drug instruction manuals for pharmacists prepared by pharmaceutical companies at the request of the Japan Hospital Pharmaceutical Association. The documents were edited to be concise so that users could obtain the DI efficiently. The documents consist of four main categories: (1) formulation property (106 documents); (2) medical news (the outlines of mechanisms and characteristics regarding new drugs and the latest information on medical care, 272 documents); (3) questions and answers (Q&A) on DI (31 documents); and (4) pharmaceutical law and reimbursement (207 documents). Each document contained a question and an answer to provide a clear overview of the topic. The documents were updated whenever there were any changes in the contents. Examples of documents on the website are shown in Table 1 (A: formulation property; B: medical news; C: Q&A on DI; D: pharmaceutical law and reimbursement).

Google Analytics and data collection: We installed Google Analytics by adding a tracking tag to the SAGASU-DI in October 2016. The tag allows for the collection of various types of data about user behavior when visiting our website. The information is summarized in a real-time, interactive dashboard format, which can be accessed by logging in. The Google Analytics data are displayed as aggregated and do not contain any personally identifiable data, making it an accessible tool without ethical concerns [11].

We collected data comprising the following metrics: the number of visits (*n*), the total number of page views (*n*), mean browsing time (seconds), bounce rate (%), job category (*n*), access methods (*n*), the type of devices used to access, and the time of day the site was visited. The bounce rate was defined as the percentage of only a single page visit during a session. The questionnaire on the job category of viewers was displayed before browsing the website documents. The types of jobs included “hospital pharmacist”, “community pharmacist”, “other pharmacist”, “practitioner”, “hospital doctor”, “dentist”, “nurse”, “medical student”, “others” and “non-medical staff”. Access methods were classified as direct links (typing the web URL directly into a browser), organic search (entry through a search engine), referrals through another website, and social media.

Time to display the pop-up window: We investigated the time during which a pop-up window was displayed for viewers to answer the pop-up questionnaire. Since the pilot investigation indicated that the actual time to browse the SAGASU-DI was 26, 27, and 29 s per month over the last three months, the pop-up window was set to 20, 25, or 30 s after viewing the document every two days. Then, we determined the optimal time taken to show the pop-up window.

Pop-up questionnaire: The pop-up questionnaires were as follows: “It was useful for my question”, “It was not useful for my question”, and “I could not read the whole text”, to evaluate the usefulness of the SAGASU-DI to viewers. The viewers were able to avoid answering the questionnaire by clicking the “Cancel” button in the pop-up window or by clicking outside the window.

The protocol to construct the SAGASU-DI and the flowchart of the user survey are shown in Figure 1.

## 3. Results

### 3.1. Overall Engagement

The main characteristics of the SAGASU-DI data obtained from 1 May to 31 July are reported in Table 2. The total number of visits and page views increased per month for three months. An average of 25,447 users visited our website, and community pharmacists accounted for 40.2% (Table 3). On average, 33,000 pages were viewed, with an average browsing time of 29 s. The average bounce rate was 90.0%. Of the total 99,000 pages viewed on the website, 49,241 (49.7%) consisted of the following 30 documents: 15 documents related to pharmaceutical law and reimbursement, 10 to formulation properties, 3 to Q&A on DI, and 2 to medical news. Organic search accounted for the highest proportion (mean, *n* = 24,200, 94.8%) of all visits to our website, followed by direct link (mean, *n* = 1042, 4.1%), referral from other websites (mean, *n* = 272, 1.1%), and social media (mean, *n* = 16, 0.06%). Our website was accessed mostly through desktop (mean, *n* = 13,199, 51.9%) and mobile (mean, *n* = 11,266, 44.3%) systems. The number of users that accessed the website based on the day of the week and time of day are shown in Figure 2 (day of the week, Figure 2a; time of day, Figure 2b). At the median, over 1000 users visited the site on weekdays, and more than 60 users visited the site from 9 a.m. to 5 p.m.

### 3.2. Time to Display Pop-Up Window

The results of investigating the appropriate time taken to display the pop-up window are shown in Table 4. The number of page views had no differences between the three times (20 s, *n* = 2497; 25 s, *n* = 2340; 30 s, *n* = 2394); however, the number of displays decreased as the time to display the pop-up window increased (20 s, *n* = 1061; 25 s, *n* = 912; 30 s, *n* = 894). The percentage of each answer to the pop-up questionnaire did not show any difference between the three times. Based on this finding, the evaluation of the pop-up questionnaire was set to 20 s after the documents were displayed.

### 3.3. Evaluation of the Utility of the SAGASU-DI

A survey on the satisfaction of users is shown in Table 5. Of the total number of page views for 3 months, 47.4% of viewers answered the pop-up questionnaire. On average, 23.2% of viewers were satisfied with our website, and 1.5% of the viewers were not. An average of 13.8% of viewers did not read the whole text, and an average of 61.5% of viewers canceled the questionnaire. Of the viewers who evaluated our website, an average of 94.1% of viewers were satisfied with our website.

## 4. Discussion

The SAGASU-DI had a total of 25,447 users, with 33,000 page views, a browsing time of 29 s on average, and a 90.0% bounce rate. Indeed, 94.1% of the viewers who evaluated our website were satisfied with it. Our findings indicated that the SAGASU-DI, which was created based on inquiries to clinical pharmacists, is valuable as a DI service.

To date, several studies have utilized Google Analytics as a tool to evaluate health websites [12,13,14,15,16,17,18,19]; however, the tool should be used with careful consideration. For example, Isenor et al. reported that a website related to DI for pharmacy students had more than 5000 users per month [20]. Moreover, a website relating to specific diseases has an average of 2441 to 9016 page views per month [15,18,19]. Compared with these websites, our website showed 25,447 users and 33,000 page views. This could probably be because our website consists of a broad DI in various medical fields. Currently, it is difficult to compare the results across various interventions because the number of users and page views are not defined [21,22]. However, there was a huge increase in the number of users and pageviews over the three months analyzed (Table 2). Therefore, our website can be considered an in-demand website for DI resources.

Since the number of drugs entering the market is increasing compared with that in the past, it is difficult to remember detailed information on drugs [23]. Currently, there are increasing opportunities for community pharmacists to receive questions about DI from patients and medical staff [24,25]. In a few surveys, it has been reported that community pharmacists often access DI centers through the Internet [25,26]. In this study, half the 99,000 total pages viewed were questions commonly asked in clinical practice regarding pharmaceutical law, insurance reimbursement, and formulation properties. In addition, the users, including 40% of community pharmacists, visited our website using a desktop or mobile device throughout their working hours on weekdays. It can be assumed that users retrieve information from our website to answer questions from patients or medical staff during working hours.

It is difficult to establish the criteria for website efficacy since various periods of time are used for the evaluation of websites serving different purposes. However, some studies have used the bounce rate and browsing time to evaluate the efficacy of websites [17,27,28]. On our website, the bounce rate and browsing time were 90% and 29 s, respectively. Users often need to interact with web pages to obtain the information they need; hence, there are some cases with low bounce rates. However, a high bounce rate could mean that users exited the website because they had immediately found what they were looking for, but it could also mean that users exited the site because it did not meet their expectations [19]. Moreover, there are some limitations with browsing time as a means of evaluation because it allows for multiple interpretations. For example, a large number of pages could result in an increased browsing time [19]. Thus, the investigation of only the bounce rate and browsing time is insufficient to clarify the utility of our website. Therefore, we utilized the pop-up questionnaire. The results showed that 94.1% of the viewers who answered the questionnaire were satisfied with our website. One of the main reasons for this could be that the SAGASU-DI documents were prepared based on inquiries to clinical pharmacists. This result indicated that the SAGASU-DI is a website that pharmacists can help to find DI quickly in clinical practice. However, this study might have a selection bias since the response rate to the pop-up questionnaire was 38.5%.

Our study is the first investigation to focus on the usefulness of a website for the services of pharmacists; however, some limitations of the study need to be considered. First, the data collection via Google Analytics could be slightly inaccurate because a new client ID is given every time, and the user may delete browser cookies, switches devices, or use a different browser [28,29,30]. Second, 61.5% of viewers did not respond to the pop-up questionnaire. One of the reasons may probably be that viewers browsed our website during busy working times and did not have enough time to answer the question. In the medical literature, the response rates for pop-up and web survey questionnaires have generally been reported to be 4.3% and 23.4%, respectively [12,26]. In our study, 47.4% of viewers answered the pop-up questionnaires, which is much higher than past data; therefore, it seems that the response rate is enough to consider the opinions of viewers. Third, users may be Japanese primarily because the website was created in the Japanese language. However, in this study, the users, except for Japanese people, would have been counted because the automatic translation function was implemented on the website. Forth, the protocol to validate the answers was not included in the construction process of the SAGASU-DI, and the references for the answers are listed in each Q&A document. Moreover, errors in the answers have not been pointed out until now. Finally, the number of users of this website seemed to be limited; however, in fact, a total of 61,576 people accessed the website over a three-month period, and the occupations included not only pharmacy pharmacists but also hospital pharmacists, physicians, nurses, medical office staff, medical students, etc.

## 5. Conclusions

We revealed that the SAGASU-DI created based on inquiries to clinical pharmacists is a valuable resource website for DI services. However, in this study, we were not able to conclude its usefulness for community pharmacists because the percentage of community pharmacists that visited the site was 40%. In the future, we plan to build a new website for community-pharmacist-only membership and solicit inquiry content from pharmacist members belonging to various pharmacies to evaluate its usefulness for community pharmacists.

## Figures and Tables

**Figure 1 healthcare-10-01541-f001:**
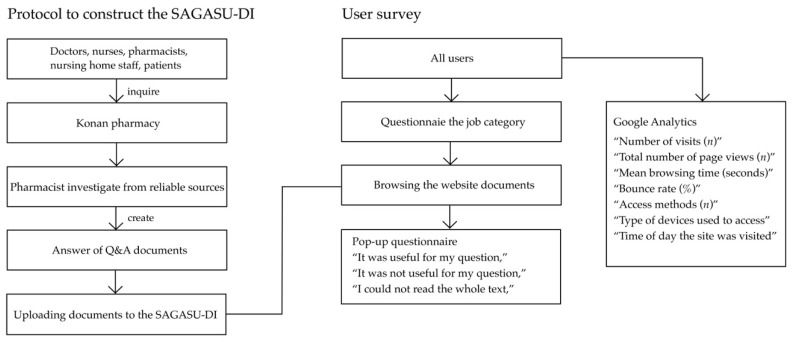
Protocol to construct the SAGASU-DI and flowchart of user survey.

**Figure 2 healthcare-10-01541-f002:**
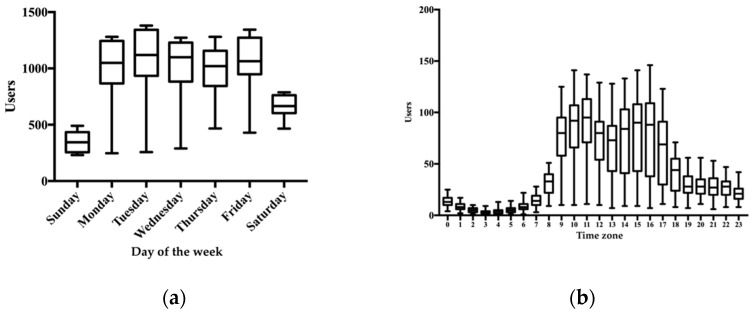
Day of week and time of day the SAGASU-DI was visited: (**a**) access counts of the SAGASU-DI relative to the day of week; (**b**) access counts of the SAGASU-DI relative to the time of day.

**Table 1 healthcare-10-01541-t001:** Examples of the website documents.

Category	Question	Answer
(1) Formulation property	What is the difference between heavy magnesium oxide and light magnesium oxide?	Most of the magnesium oxide used in medicine is heavy magnesium oxide. Heavy magnesium oxide is labeled as 5 g magnesium oxide with a volume of less than 30 mL (*The Japanese Pharmacopoeia* 17th Edition). In contrast, light magnesium oxide is magnesium oxide with a volume larger than 30 mL in 5 g. Light magnesium oxide is highly dispersive and adheres to mortar and pestle; thus, heavy magnesium oxide is mainly used in medicine. The effect of heavy magnesium oxide is slightly slower than that of light magnesium oxide.
(2) Medical news	What are the characteristics of Rybelsus^®^ (Semaglutide) Tablets (Oral glucagon-like peptide-1(GLP-1)) Summary	Indication: Type 2 diabetes mellitus Oral GLP-1 receptor agonist Ingredient: Semaglutide Q. Why is there strict dosage timing? A. Rybelsus^®^ tablets are formulated with the additive Salcaprozate sodium (SNAC), an absorption enhancer, which prevents the breakdown of the drug in the stomach and facilitates the absorption of the ingredients. Since Rybelsus^®^ is a drug that is difficult to absorb by nature and is only absorbed through the stomach, it is easily affected by food.
(3) Q&A on DI	Is Teneligliptin affected by food?	The effect of food on the AUC and the 24 h post-dose value of Teneligliptin is small, and it is expected to be effective whether it is administered on an empty stomach or after a meal, so it is unlikely to be affected by food. Therefore, the drug should be taken once a day at a fixed time. (Citation: Tenelia^®^ (Teneligliptin) Tablet Interview Form)
(4) Pharmaceutical law and reimbursement	What are the restrictions on the number of prescription days for Belsomra^®^ (Suvorexant)	Belsomra^®^ (Suvorexant) does not fall under the “Cabinet Order Designating Narcotic Drugs, Narcotic Plants, Psychotropic Drugs, and Psychotropic Substances” in Japan; therefore, there is no specific limit to the number of days Belsomra^®^ (Suvorexant) can be prescribed. Eszopiclone, Ramelteon, and Rilmazafone also have no restrictions on the number of days of prescription.

Rybelsus^®^, Novo-Nordisk, Plainsboro Township, NJ, USA; Tenelia^®^, Mitsubishi Tanabe Pharma Corporation, Tokyo, Japan; Belsomra^®^, Merck, Darmstadt, Germany.

**Table 2 healthcare-10-01541-t002:** Data collection using Google Analytics.

Item		May 2021	June 2021	July 2021	Mean
Number of visits (*n*)		19,962	26,056	30,324	25,447
Total number of pages views (*n*)		26,176	34,106	38,718	33,000
Mean browsing time (seconds)		28	30	30	29
Bounce rate (%)		90.1	90.0	89.8	90.0
Access methods, *n* (%)					
	Organic search	18,767 (93.8%)	24,869 (95.1%)	28,966 (95.2%)	24,201 (94.8%)
	Directly	1004 (5.0%)	1009 (3.9%)	1113 (3.7%)	1042 (4.1%)
	Referral from other websites	229 (1.1%)	279 (1.1%)	309 (1.0%)	272 (1.1%)
	Social media	9 (0.04%)	4 (0.02%)	36 (0.12%)	16 (0.06%)
Type of devices used for access, *n* (%)					
	Desktop	10,878 (54.7%)	13,819 (52.9%)	14,899 (49.3%)	13,199 (51.9%)
	Mobile	8236 (41.4%)	11,345 (43.4%)	14,217 (47.1%)	11,266 (44.3%)
	Tablet	786 (4.0%)	979 (3.7%)	1098 (3.6%)	954 (3.8%)

**Table 3 healthcare-10-01541-t003:** Job category of SAGASU-DI users.

Job Category	Users (*n* = 16,158), *n* (%), May 2021	Users (*n* = 21,262), *n* (%), June 2021	Users (*n* = 24,156), *n* (%), July 2021	Mean
Community pharmacist	6881 (42.6%)	8516 (40.1%)	9326 (38.6%)	8241 (40.2%)
Hospital pharmacist	2247 (13.9%)	3212 (15.1%)	3483 (14.4%)	2981 (14.5%)
Non-medical staff	1690 (10.5%)	2552 (12.0%)	3289 (13.6%)	2510 (12.2%)
Others	1394 (8.6%)	1946 (9.2%)	2261 (9.4%)	1867 (9.1%)
Medical clerk	1144 (7.1%)	1312 (6.2%)	1504 (6.2%)	1320 (6.4%)
Nurse	839 (5.2%)	1198 (5.6%)	1458 (6.0%)	1165 (5.7%)
Hospital doctor	760 (4.7%)	901 (4.2%)	1005 (4.2%)	889 (4.3%)
Practitioner	460 (2.8%)	476 (2.2%)	571 (2.4%)	502 (2.4%)
Other pharmacist	404 (2.5%)	542 (2.5%)	634 (2.6%)	527 (2.6%)
Medical student	292 (1.8%)	518 (2.4%)	523 (2.2%)	444 (2.2%)
Dentist	47 (0.3%)	89 (0.4%)	102 (0.4%)	79 (0.4%)

**Table 4 healthcare-10-01541-t004:** Answers from viewers to the pop-up questionnaire for two days in the three groups relative to the timing of displaying the pop-up window.

Item	20 s	25 s	30 s
Page views (*n*)	2497	2340	2394
Display (*n*)	1061	912	894
“It was useful to my question”, *n* (%)	279 (26.3%)	233 (25.5%)	249 (27.9%)
“It was not useful to my question”, *n* (%)	9 (0.8%)	13 (1.4%)	9 (1.0%)
“I could not read the whole text”, *n* (%)	143 (13.5%)	114 (12.5%)	126 (14.1%)
“Cancel”, *n*(%)	630 (59.4%)	552 (60.5%)	510 (57.0%)

**Table 5 healthcare-10-01541-t005:** Answers from viewers to the pop-up questionnaire for 3 months.

Item	Views (*n* = 12,315), *n* (%), May 2021	Views (*n* = 16,349), *n* (%), June 2021	Views (*n* = 18,276), *n* (%) July 2021	Mean
“It was useful to my question”, *n* (%)	3091 (25.1%)	3809 (23.3%)	4008 (21.9%)	3636 (23.2%)
“It was not useful to my question”, *n* (%)	150 (1.2%)	265 (1.6%)	274 (1.5%)	230 (1.5%)
“I could not read the whole text”, *n* (%)	1575 (12.8%)	2365 (14.5%)	2539 (13.9%)	2160 (13.8%)
“Cancel”, *n* (%)	7499 (60.9%)	9910 (60.6%)	11,455 (62.7%)	9621 (61.5%)

## Data Availability

Not applicable.

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
