# Peer review of "Usefulness of a Drug Information Resources Website (SAGASU-DI) Created Based on Inquiries to Clinical Pharmacists"

_healthcare, 2022, doi:10.3390/healthcare10081541_

Round 1

Reviewer 1 Report

The manuscript is oriented to have the usefulness of a drug information resource for clinical pharmacists

It is always important to have drug information sources that are easy to manage and gives sound and fast information in order to solve problems

The work is interesting as it has used Google Analytics in order to know if this source is of utility for health care professionals, and pharmacists in particular.

One think that I consider that should be better explained in the text is how has been constructed the SAGASU-DI. From where the information is compiled, how is it validated? 

Do the authors know what was the kind of question more consulted?

Author Response

Manuscript ID: healthcare-1790655

Title: Usefulness of a drug information resources website (SAGASU-DI), created based on inquiries to clinical pharmacists

Article type: Article

healthcare

Thank you for your reviewing our article. We made some changes in our manuscript according to reviewer’s suggestions with yellow highlight. We think some revises enhanced the quality of our manuscript.

Reviewer #1:

  1. One think that I consider that should be better explained in the text is how has been constructed the SAGASU-DI. From where the information is compiled, how is it validated?

Response: Thank you for your comment. The questions of Q&A documents on the website have been created based on inquiries about the drug information (DI) that received by Konan Pharmacy from doctors, pharmacists, nurses, nursing home staffs and patients. The answers of Q&A documents have been created based on reliable information sources such as “prescribing information” which are official documents in accordance with the Pharmaceutical Affairs Law and the interview forms which are drug instruction manual for pharmacists prepared by pharmaceutical companies at the request of the Japan Hospital Pharmaceutical Association. Although the protocol to validate the answers is not included in this process to construct the SAGASU-DI, the references for the answers are listed in each Q&A document. Moreover, errors for the answers have not been pointed out until now. The above sentences were added in the sections of Method and Limitation (p2 L72-79, p8 226-229). And, the protocol to construct the SAGASU-DI was added as Figure 1.

  1. Do the authors know what was the kind of question more consulted?

Response: Thank you for your comment. Of course, the authors understand what the questions were. Of the total 99,000 pages viewed on the website, 49,241 (49.7%) consisted of the following 30 documents: 15 documents related to Pharmaceutical law and reimbursement, 10 to Formulation property, 3 to Q&A on DI, and 2 to Medical news. In this study, half of the 99,000 total pages viewed were questions commonly asked in clinical practice regarding pharmaceutical law, insurance reimbursement, and formulation properties. The above sentences were added in the sections of Results and Discussion (p4 L132-135, p8 187-190) .

Reviewer 2 Report

I'm pleased to review this manuscript.

This manuscript is about the authors' introduction and evaluation of their DI source.

This DI source may be useful for particular people but the manuscript appears to be just a report on site usage. The number of interested people seems to be limited.

Tables and figures are not sufficient to demonstrate the scientific utility of this site (60% of visitors did not answer to questionnaire).

Q and A examples in Table 1, most questions are quite general and not information that can only be obtained from this site.

Overall, this manuscript lacks a scientific perspective in its discussion.

I evaluated that this manuscript is not in a form that can be adopted as a regular article in Healthcare journal.

I recommend submitting this manuscript to other journals as a short communication or letters.

Author Response

Manuscript ID: healthcare-1790655

Title: Usefulness of a drug information resources website (SAGASU-DI), created based on inquiries to clinical pharmacists

Article type: Article

healthcare

Thank you for your reviewing our article. We made some changes in our manuscript according to reviewer’s suggestions with yellow highlight. We think some revises enhanced the quality of our manuscript.

Reviewer #2:

  1. This manuscript is about the authors' introduction and evaluation of their DI source.

This DI source may be useful for particular people but the manuscript appears to be just a report on site usage. The number of interested people seems to be limited.

Response: Thank you for your comment. As noted, the number of users of this website seemed to be limited, however in fact, a total of 61,576 people accessed the website over a three-month period, and the occupations included not only pharmacy pharmacists, but also hospital pharmacists, physicians, nurses, medical office staff, medical students, and many others. This study was conducted to evaluate the status of use and usefulness of this website, including above-mentioned items.

The above sentences were added in the section of Limitation (p8 L229-232).

  1. Tables and figures are not sufficient to demonstrate the scientific utility of this site (60% of visitors did not answer to questionnaire).

Response: Thank you for your comment. In the medical literature, the response rates for pop-up and web surveys questionnaires have generally been reported to be 4.3% and 23.4%, respectively [12,26]. In our study, 47.4% of viewers answered the pop-up questionnaires, which is much higher than past data, therefore it seems that the response rate is enough to consider the opinions from viewers. The above sentences were added in the section of Limitation (p8 L219-223).

  1. Q and A examples in Table 1, most questions are quite general and not information that can only be obtained from this site. 

Response: Thank you for your comment. In our study, the questions of Q&A documents were created based on inquiries about the DI received at Konan Pharmacy, and the answers were created based on the information sources such as the prescribing information and interview forms. As you noted, the Q and A examples in Table 1 are quite general. However, there are few centralized websites for drug information, and our website would make it easier to obtain reliable information in the clinical practice.

  1. Overall, this manuscript lacks a scientific perspective in its discussion.

Response: Thank you for your comment. Some changes in our manuscript according to reviewer’s suggestions were made. We think that these revisions improved the quality of our manuscript and made it more informative for readers.
